# Evaluation of Stress Distribution and Force in External Hexagonal Implant: A 3-D Finite Element Analysis

**DOI:** 10.3390/ijerph181910266

**Published:** 2021-09-29

**Authors:** Vinod Bandela, Ram Basany, Anil Kumar Nagarajappa, Sakeenabi Basha, Saraswathi Kanaparthi, Kiran Kumar Ganji, Santosh Patil, Ravi Kumar Gudipaneni, Ghazi Sghaireen Mohammed, Mohammad Khursheed Alam

**Affiliations:** 1Department of Prosthetic Dental Sciences, Fixed Division, College of Dentistry, Jouf University, Sakaka 72341, Saudi Arabia; dr.mohammed.sghaireen@jodent.org; 2Department of Prosthodontics, S.V.S Institute of Dental Sciences, Appannapally, Mahbubnagar 509002, India; ram.basany@gmail.com; 3Department of Oral Surgery and Maxillofacial Diagnostics, College of Dentistry, Jouf University, Sakaka 72341, Saudi Arabia; dr.anil.kumar@jodent.org; 4Department of Community Dentistry, Faculty of Dentistry, Taif University, Taif 21974, Saudi Arabia; sakeena@tudent.edu.sa; 5Department of Pedodontics and Preventive Dentistry, St. Joseph’s Dental College and Hospital, Eluru 534004, India; sarayudoc@gmail.com; 6Department of Preventive Dentistry, Periodontics Division, College of Dentistry, Jouf University, Sakaka 72341, Saudi Arabia; dr.kiran.ganji@jodent.org; 7Department of Periodontics, Sharad Pawar Dental College, Datta Meghe Institute of Medical Sciences, Nagpur 442001, India; 8Department of Oral Medicine and Radiology, New Horizon Dental College and Research Institute, Bilaspur 495001, India; dr.santosh.patil@jodent.org; 9Department of Preventive Dentistry, Pediatric Dentistry Division, College of Dentistry, Jouf University, Sakaka 72341, Saudi Arabia; dr.ravi.gudipaneni@jodent.org; 10Department of Preventive Dentistry, Orthodontics Division, College of Dentistry, Jouf University, Sakaka 72341, Saudi Arabia; dr.mohammad.alam@jodent.org; 11Department of Dental Research Cell, Saveetha Dental College and Hospitals, Saveetha Institute of Medical and Technical Sciences, Chennai 600070, India

**Keywords:** finite element analysis, implant, stress, force, external hex implants

## Abstract

Purpose: To analyze the stress distribution and the direction of force in external hexagonal implant with crown in three different angulations. Materials and Methods: A total of 60 samples of geometric models were used to analyze von Mises stress and direction of force with 0-, 5-, and 10-degree lingual tilt. Von Mises stress and force distribution were evaluated at nodes of hard bone, and finite element analysis was performed using ANSYS 12.1 software. For calculating stress distribution and force, we categorized and labeled the groups as Implant A1, Implant A2, and Implant A3, and Implant B1, Implant B2, and Implant B3 with 0-, 5-, and 10-degree lingual inclinations, respectively. Inter- and intra-group comparisons were performed using ANOVA test. A *p*-value of ≤0.05 was considered statistically significant. Results: In all the three models, overall maximum stress was found in implant model A3 on the implant surface (86.61), and minimum was found on model A1 in hard bone (26.21). In all the three models, the direction of force along three planes was maximum in DX (0.01025) and minimum along DZ (0.002) direction with model B1. Conclusion: Maximum von Mises stress and the direction of force in axial direction was found at the maximum with the implant of 10 degrees angulation. Thus, it was evident that tilting of an implant influences the stress concentration and force in external hex implants.

## 1. Introduction

Implant therapy has emerged as an extensively accepted treatment modality during the past few decades with a reported long-term success rate ranging from 79.5% to 100% [1,2,3]. Since their introduction, osseointegrated dental implants have been used all over the world for treating patients in partially and completely edentulous conditions. Endosseous dental implants are used to retain and/or support prosthesis for a variety of tooth loss situations [3]. Brånemark’s historical discovery of implant began with the abutment connection [4]. Brånemark system was developed for facilitating implant insertion by characteristic external hexagon design feature rather than providing an anti-rotational effect [5]. “External hexagon” or “hex” is an external connection used as anti-rotational and for indexing feature. A hexagon is a six-sided shape used at the abutment–implant interface as an anti-rotational feature whereas external connection is a type of connection feature that extends above the coronal portion of an implant. Abutment connections can be external spline or external octagon, but external hex is the most common one [4]. This configuration has been incorporated in various implant systems and has served well over the years [5]. Originally, Brånemark’s implant was composed of 0.7 mm external hex with a butt joint. These dental implants were used in treating completely edentulous patients that were coupled with one-piece metal substructure. Thus, there was little interest in anti-rotational feature. For the facilitation of the surgical placement of implant, the external hex portion was added to the design [4].

There are advantages as well as disadvantages with external hex connection, these being availability of long-term follow-up data and being well suited and adapted to various implant systems; moreover, if complications arise, due to their extensive use, solutions are found throughout the literature. Shortcomings are high occurrence of screw loosening, rotational misfit, compromised aesthetic result, and microbial seal being inadequate [4]. In spite of having disadvantages, due to the long-term follow-up data, external hex is still used by some manufacturers, owing to its advantages.

With recent advances in implant therapy, restoration of function and aesthetics has become an inseparable part in replacing lost teeth. This means that an increased number of forces at abutment connections are expected. This challenge has reinvigorated research in developing better forms of abutment connections. The height and width of the exterior hex design have undergone various changes [4].

With external connection, high prosthetic success can be achieved, but screw loosening is the most common prosthetic complication, especially when replacing a single tooth. The early 0.7 mm connection provided limited screw engagement since it is shorter in length and also narrow platform produces a short fulcrum arm that creates adverse tipping forces that lead to increased screw loosening. In order for the hostile force distribution and instability to be overcome, the immediate solution was to increase the height and width of abutment connection. Height of currently available external hex ranges from 0.7 to 1.2 mm and 2.0 to 3.4 mm widths, depending on the manufacturer. These improvements bring an increase of fulcrum arm and extended abutment screw engagement. As a result, the tipping forces on abutment screws are limited, and screw loosening has become less common [4].

The aim of the present study was to analyze the stress distribution and force in and around external hexagonal implant and their prosthetic crown with three different implant angulations.

## 2. Materials and Methods

Study design and characteristics: The physical and mechanical properties of the implant, abutment, and crown used in the study are listed in Table 1.

Material properties: The models used were homogenous, isotropic, and linear elastic in this study [6,7,8]. The Poisson’s ratio (ν) and Young’s modulus of elasticity (E) of materials incorporated into the model as shown in Table 2. The implant used was Nobel Speedy Groovy (Nobel Biocare, Göteborg, Sweden), an endosseous, root form, external hexagon implant with Procera abutment and Zirconia crown. A total of 3 finite element models (FEM), one for each orientation in different angulation, were constructed separately. The model was divided into smaller elements and each element was interconnected at various discrete points called nodes. The number of nodes and elements in model A were 40,342 and 227,678, in B 40,500 and 232,474, and in C 40,556 and 237,890, respectively. We applied 300 N onto the occlusal surface of mandibular first molar model [9,10,11,12,13,14]. Once the specified force was applied, the models responded accordingly and the pattern of stress distribution was displayed in the computer and, depending on the colors, the mathematical model of mandible with 0-, 5-, and 10-degree lingual inclinations on and around bone; implant interface; and also the direction of force with respect to three planes were evaluated. Von Mises stress throughout the structure was determined using the displacement of each of these nodes [6].

Finite element analysis: The arithmetical models of implant, abutment, and inner-screw were modeled using Solid Edge software, and the tooth and mandible models were obtained from computed tomography scan by Mimics software. For meshing, the geometric models were then imported to Hypermesh software. Meshing of FEM consists of nodes and elements. Assembled FEM of the tooth and implant was then imported into ANSYS 12.1 software, (Ansys, Canonsburg, PA, USA) from which results were obtained for analysis. Flowchart of study design is shown in Figure 1.

Study models: For calculating stress distribution and force, we categorized and labeled the following groups accordingly as Implant A1, Implant A2, and Implant A3, and Implant B1, Implant B2, and Implant B3 with respect to 0-, 5-, and 10-degree lingual inclinations, respectively. Implants A1 and B1 were made parallel to the long axis of the bone model, whereas implants A2 and B2 had a 5 degree inclination lingually, and implant models A3 and B3 had a lingual inclination of 10 degrees.

Sample size and statistical analysis: Using G-power computing tool, we used the effect size thus obtained to determine the samples for each group. Statistical Package for Social Sciences (SPSS) version 24 (IBM. Inc., Chicago, IL, USA) was used for statistical analysis. To describe mean values of stress and direction of forces, we utilized descriptive statistics. Inter- and intra-group comparison were performed using ANOVA test. A *p*-value of ≤0.05 was considered statistically significant. Using the Scheffe test, post hoc intra-group comparisons were evaluated.

## 3. Results

In the models A1, A2, and A3, the distribution of von Mises stress values resulted from vertical loading on the crown, interface of bone-implant, and cortical bone surrounding the implant. In models B1, B2, and B3, the direction of force along all the three axes was measured.

### 3.1. Von Mises Stress in Models A1, A2, and A3 (All Values in MPa)

In model A1, maximum stresses were found at implant body (85.39), and minimum stress was on hard bone (26.12) with medium on the crown (70.35) (Figure 2). In model A2, maximum stresses were found at implant body (73.59) and minimum stress on hard bone (40.25), with medium stress on the crown (72.67) (Figure 3). In model A3, the maximum stress was found on the implant (86.61) and minimum on the hard bone (51.86), with medium stress on crown part (76.25) (Figure 4). In all the three models, overall maximum stress was found in model A3 on the implant surface (86.61), and minimum was found on model A1 in hard bone (26.21), as shown in Table 3.

### 3.2. Direction of Force on Models B1, B2, and B3

The direction of force was also studied in three planes, i.e., mesio-distal directions (DX), vertical/axial direction (DY), and bucco-lingual direction (DZ). In model B1, the maximum force was along the direction of DY (0.0151), minimum with DZ (0.002), and medium force along DX (0.01025) (Figure 5). In model B2, maximum force was along the direction of DY (0.0139), minimum along DZ (0.006), and medium force along DX (0.0048) (Figure 6). In model B3, maximum force was along DY (0.0175) and minimum in DX (0.0017), with medium force along DZ (0.0109) (Figure 7). In all the three models, the direction of force along the three planes was maximum in DX (0.01025) and minimum along DZ (0.002) direction with model B1 only (Table 4).

## 4. Discussion

Various factors that influence the stress distribution around the implant supporting prosthesis and the adjacent bone include the type of load; bone–implant interface; length and diameter of the implant; the connection type, shape, and characteristic features of the implant surface; the type of prosthesis; and the quantity and quality of the surrounding bone [15]. In the present study, three different models with the external hex fixture-abutment connection of 0-, 5-, and 10-degree angulations and direction of force in *X*-, *Y*-, and *Z*-axes were made, simulating natural conditions. The FEM was held rigidly at the distal ends and also at the bottom to allow some amount of bending in the mandible [16]. It is obvious that the model was only a rough estimate of clinical situation. Therefore, focusing on the qualitative comparison for analysis will be sensible as opposed to on quantitative data. FEM was generated using Hypermesh 9.0 software, and ANSYS software version 12.1 was used for a finite element programming. A mathematical model of mandible with average length (13 mm) and diameter (5 mm) implant, abutment, and prosthesis with average masticatory loads (300 N) was utilized according to in vitro protocol for the study as per experiments and suggestions by various researchers [6,10,16,17,18]. The implant used was made of titanium–aluminum–vanadium (Ti 6A1 4 V) [19,20].

Model Considerations and Loading conditions: By using CT scan, we generated the mechanical model of a partially edentulous mandible, considering missing first molar, as it provides the exact bony contours of the bone [21]. In the current study, mandibular first molar was considered as this is the most common missing tooth in accordance with studies performed by previous authors [9,10,11,12,13,14]. To simulate muscle forces, we applied the boundary conditions at the mesial end of the second premolar and distal end of the second molar of the mandible [22]. The muscle forces used were static forces rather than dynamic in the model, which is in accordance with previous studies [23,24]. Elastic modulus and Poisson’s ratio were considered for structuring the model of implant, abutment, and crown. Cortical, cancellous bone and implant with superstructure were assumed to be linear elastic, homogenous, and isotropic [22,25]. The implant abutment connection plays an important role in direction of stress from crown to implant and eventually to the surrounding bone.

Von Mises stresses are the most commonly reported stress in FE studies in terms of assessing the overall stress. Any excess stress (compressive and tensile) can lead to bone resorption and eventual necrosis. By comparing the corresponding stress, the factors that may lessen the potential harmful stress can be investigated. Threaded implants increase the surface area for compressive loading in proportion to the number and depth of the threads, which are also able to transmit axial loads to the surrounding bone by compression on the faces of the threads [11]. The close interlocking at the microscopic plus the macroscopic form of the screw allows for load transfer without any tendency for slippage. The design of the implant also influences the stress concentration around various parts and the interface. Another important factor in the stress pattern determination is the type of load under consideration, angulation of the implant, and the manner in which they are transferred to the surrounding tissues [20]. Moreover, wide-neck implant design showed a low probing depth and less marginal bone loss when compared to the narrow-neck implant [26]. The deeper the implant placed in the bone, the less the stress concentrations in comparison with the superficially placed implant [27]. If the force is compressive or vertical and is transferred uniformly through the implant body, the load-bearing area would be the apical base of the implant, whereas in a shear/oblique type of force, the sides of the implant are loaded [22]. The abutment should be screwed following manufacturer recommendation for appropriate torque. This is to avoid the vertical discrepancy that might modify the height of the abutment, leading to a worse stress distribution [28].

Two- (2-D) and three (3-D)-dimensional FEA techniques have been used extensively in dentistry. The clinical situation cannot be validated as the horizontal and/or oblique bite forces are not possible to study with 2-D FEA [29]. Hence, a 3-D FEA is preferred it is an actual representation of stress behavior on supporting bone. In constructing a 3-D model, it is important to simulate the real clinical situations. In some previous studies, the boundary condition that is rigid was used [6,30]. In this study, a 3-D FEA was generated to know the exact biomechanical behavior of the supporting hard tissues of the mandible.

Limitations of this study are that certain assumptions were made in material properties, boundary and geometric considerations, and bone–implant interface. Moreover, there is an inability to simulate complex biological processes.

The clinical relevance of the study is that while placing the implants in the osteotomy, one should consider the implant angulation and the implant design, as well as the bone morphology.

## 5. Conclusions

Maximum von Mises stress and force in vertical/axial direction was found in the implant with external hexagonal design with an angulation of 10 degrees. Implant angulation influences the stress distribution and the amount of force falling on implants. Thus, an implantologist should keep in mind the angulation during osteotomy preparation for successful implant osseointegration and prosthesis.

## Figures and Tables

**Figure 1 ijerph-18-10266-f001:**
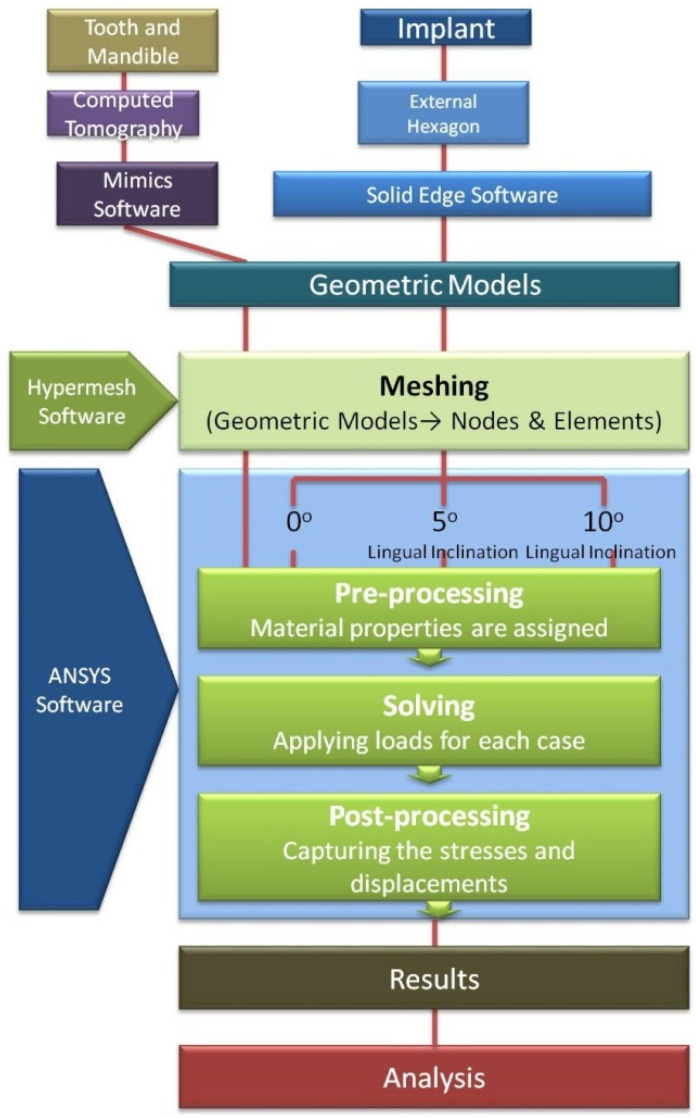
Flowchart of entire study.

**Figure 2 ijerph-18-10266-f002:**
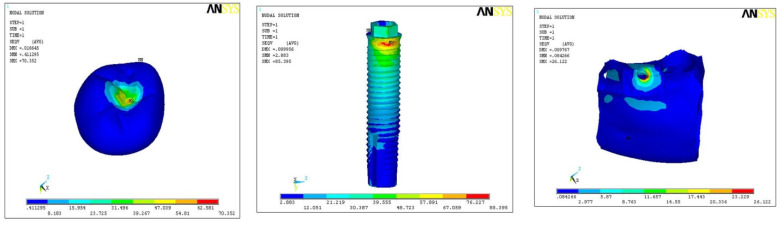
Pattern of stress in model A1.

**Figure 3 ijerph-18-10266-f003:**
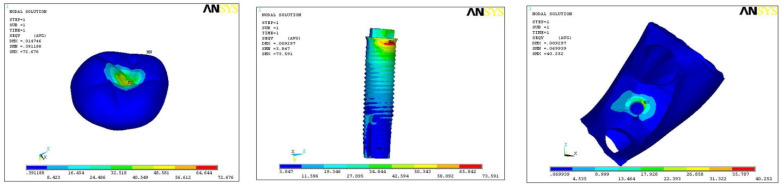
Pattern of stress in model A2.

**Figure 4 ijerph-18-10266-f004:**
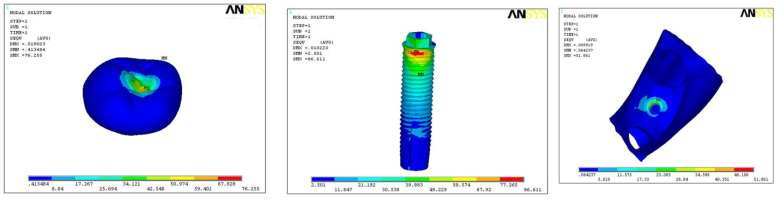
Pattern of stress in model A3.

**Figure 5 ijerph-18-10266-f005:**
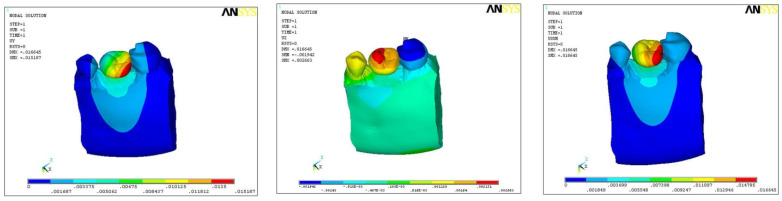
Direction of force in model B1.

**Figure 6 ijerph-18-10266-f006:**
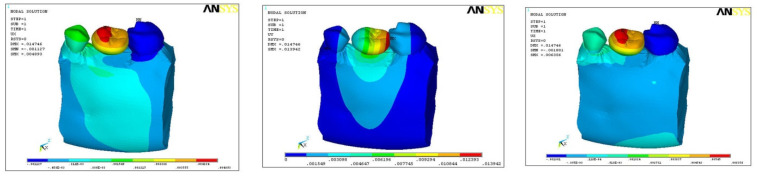
Direction of force in model B2.

**Figure 7 ijerph-18-10266-f007:**
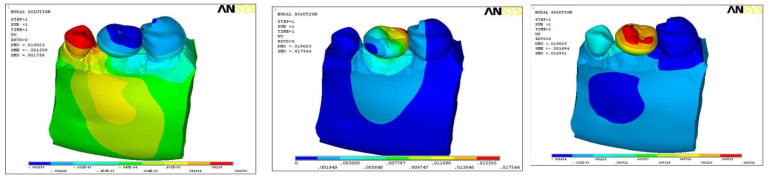
Direction of force in model B3.

**Table 1 ijerph-18-10266-t001:** Physical and mechanical properties of implant, abutment, and crown.

Length of Implant	13 mm
Diameter of implant	5 mm
Length of abutment	6.5 mm
Diameter of abutment	5 mm
Yield strength	760 MPa
Tensile strength	860 MPa
Flexural strength of crown	1120 MPa

**Table 2 ijerph-18-10266-t002:** Material properties used in preparing the finite element model.

Material	E (MPa)	Nu (ν)
Dentine	18,600	0.31
Hard bone	15,000	0.33
Soft bone	1500	0.3
Periodontal ligament	50	0.45
Implant	110,000	0.35
Abutment	114,000	0.34
Inner screw	205,000	0.33
Crown	70,000	0.19

**Table 3 ijerph-18-10266-t003:** Von Mises stress values in all models.

*n* = 14		Von Mises Stress (N/M^2^)			ANOVA	
		Crown	Implant	Hard Bone	F-Value	*p*-Value
Model A 1	0 Degree	70.35 (±14.85)	85.39 (±22.65)	26.12 (±8.95)	25.8	0.00 *
Model A 2	5 Degree	72.67 (±11.25)	73.59 (±16.78)	40.25 (±11.87)		0.00 *
Model A 3	10 Degree	76.25 (±18.98)	86.61 (±14.68)	51.86 (±10.57)		0.00 *

* Statistically significant (*p* < 0.05).

**Table 4 ijerph-18-10266-t004:** Direction of force in all models.

*n* = 14		Direction of Force (N)			ANOVA	
		*X*-Axis	*Y*-Axis	*Z*-Axis	F-Value	*p*-Value
Model B 1	0 Degree	0.01025 (±0.004)	0.0151 (±0.003)	0.002 (±0.003)	0.001	0.012
Model B 2	5 Degree	0.0048 (±0.001)	0.0139 (0.006)	0.006 (±0.004)		0.02
Model B 3	10 Degree	0.0017 (±0.008)	0.0175 (±0.005)	0.0109 (±0.008)		0.04

## Data Availability

The data presented in this study are available on request from the corresponding author.

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
