# Peer review of "Evaluation of Stress Distribution and Force in External Hexagonal Implant: A 3-D Finite Element Analysis"

_ijerph, 2021, doi:10.3390/ijerph181910266_

Round 1

Reviewer 1 Report

Dear Authors
Evaluation of stress distribution and force in external hexagonal 2 implant: A 3-D finite element analysis. I reviewed the article. Different types of implant abutment connections were evaluated with the 3-D finite element model. This study will make a positive contribution to dentists who apply dental implants. I just detected an error “Figure 4. Pattern of stress in model A2.” I think the correct sentence would be “Figure 4. Pattern of stress in model A3

Author Response

Respected Reviewer,

Necessary corrections were made as per your recommendation. The legend name for figure-4 has been updated.

Reviewer 2 Report

Dear authors,

present manuscript has merit although it can be improved.

Discussion, line 152

When talking about factors that influences stress distribution around implants all that was written is correct but please cite some more works related to those factors.

For example, different implant neck design might also have an impact 

please cite PubMed ID: 32106401

Another example could be different  depths of implant positioning

please cite PubMed ID: 26468803

Discussion 164

When talking about the abutment of the mathematical model, please specify that the abutment should be screwed following manufacturer recommendation regarding appropriate torque.

This is to avoid a vertical discrepancy that might modify the height of the abutment thus leading to a worse stress distribution.

Please cite << Vertical discrepancy in height of morse-cone abutments submitted to different torque forces. Materials 2021, 14,4950. >> 

Please cite PubMed ID 20924534.

Author Response

Respected Reviewer,

Thank you for reviewing the article and providing us with your valuable comments.

Necessary corrections were made as per your recommendation. The manuscript has been updated and citations were added and highlighted as per your suggestion.

The point to point clarification was addressed using table in the attached word file that has been uploaded as attachment.

Thank You,

Regards

Dr. Vinod Bandela

Reviewer 3 Report

Study design explanation could be improved. 

Sample groups A1, A2, A3 and B1, B2, B3 are not clearly explained.

Methods used and design of samples to measure stress forces could be clearly explained.

Limitations of the study and clinical relevance is not explained

Author Response

Respected Reviewer,

Thank you for reviewing the article and providing us with your valuable comments.

Necessary corrections were made as per your recommendation. The manuscript has been updated and highlighted with suggested correction.

The point to point clarification was addressed using table in the attached word file that has been uploaded as attachment.

Thank You,

Regards

Dr. Vinod Bandela

Reviewer 4 Report

The information provided as a basis for the introduction is sufficient and very well explained. The construction of the introduction itself and the data provided is sufficient. One suggestion I could make is that by stating:

"Implant therapy emerged as an extensively accepted treatment modality during the 26 past few decades with a reported long-term success rate ranging from 79.5% to 100%. "

Usually, when it comes to percentages you could certainly find a study to cite in terms of long-term success rates in implant therapy.

Everything is presented correctly in the results. You will need to change the displayed images instead. These are Figures 1,2,3,4,5,6,7. They are not loaded at maximum clarity.

I say this because the details at the bottom of the image as well as those at the top left of the image cannot be read. You will need to save them to a larger size to provide better clarity and visibility.

Author Response

Respected Reviewer,

Thank you for reviewing the article and giving your valuable suggestions.

Necessary corrections have been done in the manuscript and highlighted. 

Point to point clarification has been provided as a table in the word file that has been uploaded as attachment.

Thank you,

Regards,

Dr. Vinod Bandela
